# Freeze-Dried Tuna Pepsin Powder Stabilized by Some Cryoprotectants: In Vitro Simulated Gastric Digestion toward Different Proteins and Its Storage Stability

**DOI:** 10.3390/foods11152292

**Published:** 2022-08-01

**Authors:** Umesh Patil, Mehdi Nikoo, Bin Zhang, Soottawat Benjakul

**Affiliations:** 1International Center of Excellence in Seafood Science and Innovation, Faculty of Agro-Industry, Prince of Songkla University, Hat Yai 90110, Songkhla, Thailand; umesh.p@psu.ac.th; 2Department of Pathobiology and Quality Control, Artemia and Aquaculture Research Institute, Urmia University, Urmia 57179-44514, West Azerbaijan, Iran; m.nikoo@urmia.ac.ir; 3College of Food and Pharmacy, Zhejiang Ocean University, Zhoushan 316022, China; zhangbin@zjou.edu.cn

**Keywords:** pepsin, extraction, enzyme stability, protein digestion, storage

## Abstract

The impact of maltodextrin (10%) in combination with trehalose or glycerol at different levels (2.5% and 5%) and their mixture on the stability of freeze-dried pepsin from skipjack tuna stomach was studied. Addition of 5% trehalose and 10% maltodextrin yielded the powder (TPP-T5) with highest relative pepsin activity (*p* < 0.05). TPP-T5 had different shapes and sizes, with mean particle size of 65.42 ± 57.60 μm, poly-dispersity index of 0.474, and zeta potential of −19.95. It had bulk density of 0.53 kg m^−3^ and possessed fair flowability. The wetting time for TPP-T5 was 16.36 ± 0.73 min, and solubility was 93.58%. TPP-T5 stored at room temperature under different relative humidities could maintain proteolytic activity up to 4 weeks. Commercial porcine pepsin (CP) and crude tuna pepsinogen had molecular weights of 35.2 and 43.3 kDa, respectively, when analyzed using gel filtration (Sephadex G-50) and SDS-PAGE. Tuna pepsin had comparable hydrolysis toward threadfin bream muscle protein, whey protein isolate, and kidney bean protein isolate to commercial pepsin, especially at a higher level (15 units/g protein). Digested proteins contained peptides with varying molecular weights as determined by MALDI-TOF. Therefore, pepsin from skipjack tuna stomach could replace commercial porcine pepsin and was beneficial supplement for patients with maldigestion, particularly the elderly.

## 1. Introduction

Digestive enzymes play a major role in the digestion of foods, particularly proteins, which are known as essential nutrients for repairing and building body tissues and coordinate several functions in the human body. In the initial phase of food digestion, the stomach produces a mixture of compounds recognized as “gastric juice”. Gastric juice mainly contains pepsinogen and hydrochloric acid (HCl) [1]. HCl decreases the pH of the stomach, resulting in the autocatalytic conversion of pepsinogen into active pepsin [2]. Pepsin is mainly involved in protein digestion by the initial and partial hydrolysis of proteins into smaller peptides and amino acids that are further hydrolyzed and absorbed in the small intestine [3]. However, modern lifestyle and aging affect the digestive system in numerous ways. Lee et al. [4] reported that pepsin secretion was decreased by around 40% in the elderly, especially those involved in smoking, and associated with *Helicobacter pylori* infection and atrophic gastritis. Reduced secretion of digestive enzymes negatively affects the ability to absorb nutrients from food [5]. Therefore, supplementing digestive enzymes is an alternative treatment for a person with enzyme deficiencies. Pepsin is commercially produced from bovine and porcine for enzyme supplementation [6,7]. Pepsin extraction condition generally determine pepsin activity differently. Jurado et al. [6] extracted pepsin from swine wastes using different extraction methods, which affected the proteolytic activity and yield of pepsin differently. Similarly, the properties of fish pepsin were also influenced by the extraction processes [2]. Nevertheless, alternative sources of pepsin are becoming more important due to religious limitations. An increasing interest in finding higher-value pharmaceuticals and biochemicals from fish byproducts, particularly enzymes, has been paid [8].

Skipjack tuna (*Katsuwonus pelamis*) is a valuable fish in Thailand, mostly for canning [9]. A significant amount of byproduct (≈0.4 million tons per year) is produced during fish processing [10]. Byproducts from tuna processing are often used as pet food or fishmeal, which have a low market value. However, fish viscera are recognized as a significant source of numerous enzymes, especially proteases with unique properties for industrial applications [9]. Skipjack tuna stomachs discarded in large amounts during processing can serve as a potential source of pepsin for further usage, particularly in the preparation of enzyme supplementation for patients facing malabsorption. Pepsin has a molecular weight of 35.5 kDa and is made up of a single polypeptide chain with 321 amino acids [8]. Essentially, it is present as pepsinogen, in which pro-segment is attached to pepsin [8]. The pepsinogen can be activated under acidic conditions. It is an aspartic endopeptidase and has broad cleavage specificity. Pepsin preferentially cleaves peptide bonds between hydrophobic amino acids and produces smaller peptides [11]. In the pH range of 2 to 5, active pepsin is stable. However, pepsin rapidly loses its activity at pH above 5 [8]. Moreover, enzymes are generally susceptible to physical instability and chemical degradation in aqueous solutions. As a result, enzymes generally lose their activity easily. Therefore, the native structure of protein/enzyme is required to be preserved, especially in the dried form. Freeze-drying is commonly employed in the commercial production of dry protein/enzyme since it can retain satisfactory activity during long-term storage [12]. Nevertheless, the freeze-drying process itself can also cause stresses on protein/enzyme that can subsequently lead to significant conformational changes in the absence of cryoprotectants [12].

Maltodextrin is a polysaccharide commonly used as a wall material in the encapsulation process. Corveleyn and Remon [13] reported that maltodextrin was able to protect lactate dehydrogenase against inactivation during freeze-drying. Moreover, sugar such as trehalose and sugar alcohol such as glycerol are well known as cryoprotectants frequently used for freeze-drying to stabilize enzyme or proteins [14]. Therefore, the addition of suitable coating and stabilizing materials into tuna pepsin extract can be a potential means to retain proteolytic activity after freeze-drying. However, no information on the impact of maltodextrin, trehalose, glycerol, or their mixtures on the stability of pepsin from tuna stomachs during freeze-drying exists. In addition, little information on hydrolysis efficacy towards proteins and the storage stability of pepsin powder has been documented. Thus, this study aimed to investigate the influence of maltodextrin and trehalose or glycerol or their combination at various concentrations on tuna pepsin activity after freeze-drying and to study the efficacy of the selected freeze-dried tuna pepsin powder on protein hydrolysis in the simulated gastric digestion system, compared to commercial porcine pepsin. The storage stability of tuna pepsin powder at room temperature under various relative humidities was also examined.

## 2. Materials and Methods

### 2.1. Chemicals and Materials

Maltodextrin, bovine hemoglobin, L-leucine, L-tyrosine, pepsin from porcine gastric mucosa, bovine serum albumin (BSA), Folin–Ciocalteu phenol reagent, and 2,4,6-trinitrobenzenesulfonic (TNBS) acid were purchased from Sigma-Aldrich (St Louis, MO, USA). Acetone, HCl, sodium sulfite, and sodium hydroxide were purchased from LabScan (RCI Labscan Ltd., Bangkok, Thailand). Trichloroacetic acid (TCA), potassium sulfate (K_2_SO_4_), sodium chloride (NaCl), magnesium chloride (MgCl_2_), and lithium chloride (LiCl) were procured from Loba Chemi (Mumbai, Maharashtra, India). Electrophoresis chemicals were acquired from BioRad (Richmond, VA, USA). Whey protein isolate was purchased from ProflexTM (Power Corp. Co., Ltd., Bangkok, Thailand). Threadfin bream (24–48 h after capture) were bought from a local market. Kidney bean protein isolate (KBPI) was prepared as described by Gulzar et al. [15] with the aid of the pH shift process.

### 2.2. Preparation of Skipjack Tuna Stomach Powder

Skipjack tuna (*Katsuwonus pelamis*) pooled internal organs were received from I-Tail Corporation Public Company Limited, Songkhla, Thailand. The sample (30 kg) was placed in a polyethylene bag and transported on ice (1:2, *w/w*) within 30 min. The stomach was separated from other internal organs, washed 10 times with 10 volumes of deionized water (DI), and stored at −20 °C. To defrost the frozen stomach, running tap water (26–28 °C) was used until the core temperature reached −2 to 0 °C. The sample was then chopped into small pieces (0.5 × 0.5 cm) and finely ground in liquid nitrogen using a National Model MX-T2GN blender (Taipei, Taiwan). The ground sample was referred to as ‘stomach powder’.

### 2.3. Preparation of Crude Pepsin Extract

Crude pepsin extract was prepared following the method of Nalinanon et al. [8]. Briefly, stomach powder was added to sodium phosphate buffer (50 mM, at pH 7.27) at a ratio of 1:50 (*w/v*). The mixture was stirred constantly for 1 h at 4 °C. Thereafter, the mixture was centrifuged for 30 min at 7700× *g* and 4 °C using a centrifuge (Hitachi Koki Co., Ltd., Tokyo, Japan) to remove tissue debris. The collected supernatant was measured for proteolytic activity.

### 2.4. Proteolytic Activity Assay

Prior to the assay, 1 M HCl was used to adjust the pH of crude pepsin extract to pH 2 for activation. The mixture was allowed to stand for 10 min at 4 °C, followed by centrifugation at 5000× *g* for 10 min and 4 °C. Acidified supernatant containing activated pepsin was collected and used as the source of active pepsin. The proteolytic activity of activated pepsin solution was assayed using hemoglobin as a substrate following the procedure of Nalinanon et al. [8].

### 2.5. Effect of Cryoprotectants on Relative Pepsin Activity after Freeze-Drying

Firstly, maltodextrin (10%, *w/v*) was dissolved in 25 mL of crude pepsin extract solution. Trehalose (2.5 and 5%, *w/v*), glycerol (2.5 and 5%, *w/v*), or mixed trehalose/glycerol (1.25:1.25 and 2.5:2.5%, *w/v*) were added to pepsin solution. The control was made in the same way without the addition of trehalose, glycerol, or their mixtures. The samples in vials were then freeze-dried using a SCANVAC CoolSafe™ freeze-dryer (CoolSafe 55, ScanLaf A/S, Lynge, Denmark). The freeze-dried sample was ground using mortar and pestle and the obtained powder was designated as ‘tuna pepsin powder; TPP’ and placed at −40 °C until further analyses. A flow chart of the crude tuna pepsin powder production process is described in Figure 1.

#### Relative Activity of TPP

Relative pepsin activity of TPP containing maltodextrin along with trehalose, glycerol, or their mixtures at different levels was determined. HCl solution (0.1 N) was added to TPP, and pH was adjusted to 2.0 and the final volume was adjusted to 25 mL. The proteolytic activity of all samples was determined as described previously.

The relative activity [5] was calculated as follows:(1)$$\mathrm{Relative}\;\mathrm{activity}\;\left(\%\right)=\frac{P_{f}}{P_{i}}\times 100$$
where *P_f_* is pepsin activity of TPP containing maltodextrin in the presence of trehalose, glycerol, or their mixtures at different concentrations. *P_i_* is the initial pepsin activity of crude pepsin extract before freeze-drying.

TPP showing the highest relative pepsin activity was selected for further study.

### 2.6. Characterization of Selected TPP

#### 2.6.1. Microstructure

Morphology of the selected TPP sample was observed using a scanning electron microscope (SEM) (Quanta 400, FEI, Eindhoven, the Netherlands). The sample was mounted on individual bronze stubs and sputter-coated with a gold layer (Sputter coater SPI-Module, West Chester, PA, USA). The samples were visualized at an acceleration voltage of 20 kV and 5–10 Pa pressure [15]. Magnifications of 500×, 1000×, and 5000× were used.

#### 2.6.2. Particle Size, Poly-Dispersity Index (PDI), and Zeta (ζ) Potential

A laser particle size analyzer (Model LS 230, Beckman Coulter, Fullerton, CA, USA) was used to determine the particle size of the selected TPP, following the procedure of Gulzar and Benjakul [15]. Selected TPP was appropriately diluted in deionized water and measured at 25 °C for PDI and ζ-potential using a PALS Zeta potential analyzer (Brookhaven Instruments Corp., Holtsville, NY, USA).

#### 2.6.3. Density and Flowability

Density and flowability of the selected TPP were determined using Carr index (*Ci*) as tailored by Gulzar and Benjakul [16]. *Ci* was calculated by the following equation:(2)$$Ci=\frac{\left(\rho t-\rho u\right)}{\rho t}\times 100$$
where *ρu* and *ρt* are bulk density (untapped) and tapped density of the powder sample, respectively. 

To calculate tapped density, the powder was filled in a graduated cylinder and mechanically tapped 50 times on the table [16]. The volume of the powder was calculated from the graduations of the cylinder after no further change in volume took place with tapping of cylinder to calculate tapped density. Untapped density was the normal bulk density of powder, which was calculated as mass per volume of powder without any tapping.

Ci values of <15, 15–20, 20–35, 35–45, and >45 were classified as very good, good, fair, bad, and very bad, respectively.

#### 2.6.4. Wettability and Solubility

Wettability of the selected TPP was determined by measuring the time to completely sink samples (5 g) in 100 mL of water [16]. To measure solubility, powder (0.1 g) was dissolved in 24.9 mL of pure water. The mixture was stirred for 15 min at 25 °C, followed by centrifugation at 3350× *g* for 20 min. Supernatant (10 g) was dried at 105 °C in an oven until a constant weight was achieved.

The solubility was calculated as follows:(3)$$\mathrm{Solubility}\;\left(\%\right)=\frac{m\times 2.5}{w}\times 100$$
where *m* and *w* are weight of dry matter in supernatant and weight of powder used, respectively.

### 2.7. Storage Stability of the Selected TPP at Different Relative Humidities (RHs)

The selected TPP (10 g) was added to 1 cm diameter glass vial and kept in a desiccator containing different saturated salt solutions with different RHs for 28 days at 30 °C. The saturated salt solutions included LiCl (11% RH), MgCl_2_ (33% RH), NaCl (75% RH), and K_2_SO_4_ (97% RH). At every 7 days, the relative proteolytic activity of all samples was assayed as described previously.

### 2.8. Characterization of Commercial and the Selected TPP

The selected TPP was subjected to characterization in comparison with commercial purified porcine pepsin powder (CP).

#### 2.8.1. SDS-Polyacrylamide Gel Electrophoresis (SDS-PAGE)

Electrophoretic patterns of the selected TPP and CP were examined by SDS-PAGE as tailored by Laemmli [17]. Sample was dissolved in 5% SDS at the ratio of 1:1 (*w/w*) and heated at 90 °C for 30 min. Protein content of a mixture was measured by the Biuret method [18], and 15 μg of protein sample was loaded onto the polyacrylamide gel (4% stacking gel; 12% running gel). Gels were separated at 15 mA per gel, followed by staining and destaining.

#### 2.8.2. Molecular Weight Distribution

The selected TPP and CP were separated by the size exclusion chromatography. TPP and CP were dissolved in phosphate buffer (PB, 20 mM and pH 7) prior to loading onto a Sephadex G-50 (Pharmacia Biotech, Uppsala, Sweden) (3.9 × 70 cm) column equilibrated with approximately two-bed volumes of PB. The sample was eluted with PB at the flow rate of 0.5 mL/min, and fractions of 3 mL were collected. The absorbance of fractions was read at 280 nm. In addition, pepsin activity was also determined for all the fractions. The molecular weight distribution of the sample was determined from the retention time of activity peaks, in comparison with those of standards including BSA (66,000 Da), soybean trypsin inhibitor (20,100 Da), vitamin B12 (1355 Da), Gly-Tyr (238.25 Da), and L-tyrosine (181.2 Da).

### 2.9. In Vitro Simulated Gastric Digestion of Different Proteins by the Selected TPP and CP

Kidney bean protein isolate, whey protein isolate, and fish mince were digested by the selected TPP and CP following the method of Natchaphol et al. [5], with slight modifications. Aforementioned protein sample (2 g) was dissolved in 10 mL of 0.1 N HCl; pH was checked and readjusted to 2.0 using 0.1 M HCl. Selected TPP or CP at the levels of 5, 10, and 15 units/g protein was added to the mixture to initiate hydrolysis. Control was made in the same way without the addition of TPP or CP. All samples were incubated at 37 °C for 1 h in an orbital shaking water bath (Memmert, D-91126, Schwabach, Germany) to mimic in vitro gastric digestion. To stop hydrolysis, hot SDS (5%) was added to the mixture at a ratio of 1:1, *v/v*, and it was kept in boiling water for 15 min. Thereafter, the mixture was then cooled, followed by centrifugation at 8500× *g* for 15 min using a centrifuge (Beckman Coulter, Allegra™ centrifuge, Palo Alto, CA, USA). The supernatant was collected and analyzed for protein patterns using SDS-PAGE. For the sample used for the degree of hydrolysis (DH) determination, the hydrolysis reaction was terminated by heating in boiling water for 15 min without addition of SDS.

#### 2.9.1. SDS-PAGE of Digests

Protein patterns of hydrolyzed samples were examined by SDS-PAGE as described previously.

#### 2.9.2. Degree of Hydrolysis (DH) of Digests

DH of samples was measured as detailed by Patil and Benjakul [19]. Briefly, 125 μL of diluted sample was mixed with 2.0 mL of 0.2 M phosphate buffer, pH 8.2, and 1.0 mL of TNBS (0.01%, *w/v*) solution. The mixture was vortexed and incubated for 30 min at 50 °C in the dark. To terminate the reaction, 2.0 mL of 0.1 M sodium sulfite were added to the reaction mixture. The mixtures were cooled at room temperature for 15 min. The absorbance was read at 420 nm using a UV-1601 spectrophotometer (Shimadzu, Kyoto, Japan), and α-amino group was expressed in terms of L-leucine. The *DH* was calculated as follows:(4)$$DH\left(\%\right)=\frac{L-L_{0}}{L_{max-}L_{0}}\times 100$$
where *L* is the number of α-amino groups of a hydrolyzed sample. *L*_0_ is the number of α-amino groups in the initial sample. *L_max_* is the total a-amino groups obtained after acid hydrolysis (6 M HCl at 100 °C for 24 h).

#### 2.9.3. Size Distribution of Digests

The molecular weight (MW) distribution of digested protein samples was evaluated by MALDI-TOF as detailed by Benjakul, et al. [20]. The Autoflex Speed MALDI-TOF (Bruker, GmbH, Bremen, Germany) mass spectrometer equipped with a 337 nm nitrogen laser was used.

### 2.10. Statistical Analysis

Complete randomized design (CRD) was used for the entire study. Three lots of samples were used to conduct the experiments in triplicate. The data were subjected to an analysis of variance (ANOVA). The mean comparison was conducted using Duncan’s multiple range test. The *t*-test was performed to compare the pairs [21]. The Statistical Package for Social Science (SPSS 23.0 for windows, SPSS Inc., Chicago, IL, USA) was used to conduct the statistical analysis.

## 3. Results and Discussion

### 3.1. Effect of Maltodextrin and Trehalose and/or Glycerol at Different Levels on Pepsin Activity after Freeze-Drying

Relative pepsin activity of TPP added with 10% maltodextrin in combination with trehalose and/or glycerol at various levels is displayed in Figure 2. The lowest relative pepsin activity was observed in the control (10% maltodextrin without trehalose and/or glycerol) (*p* < 0.05). However, relative pepsin activity significantly increased in all samples with the addition of maltodextrin combined with trehalose and/or glycerol (*p* < 0.05). Trehalose and glycerol are well known as cryoprotectants frequently used for freeze-drying to stabilize enzymes or proteins [14]. A cryoprotectant is thought to provide a hydrogen-bonding network to proteins/enzymes and to maintain the three-dimensional (3D) structure of the biomolecule such as enzymes during freeze-drying [22]. The highest relative pepsin activity was observed in TPP containing 10% maltodextrin and 5% trehalose when compared to other samples (*p* < 0.05). This is because the water molecules plausibly bound with trehalose molecules, thus providing a hydrogen-bonding network and maintaining the three-dimensional structure of pepsin or pepsinogen during freeze-drying. Santagapita [23] reported that trehalose stabilized invertase during freeze-drying. Essentially, trehalose has a higher number of hydrogen bond donors than glycerol. Therefore, trehalose plausibly interacted with pepsin or pepsinogen as well as water more effectively, thus stabilizing the aforementioned molecules during freeze-drying more potentially. Relative pepsin activity was augmented with increasing concentrations of trehalose and glycerol. A higher amount of trehalose and glycerol might contribute to a higher stabilizing effect toward pepsin or pepsinogen. However, a combination of trehalose and glycerol was not particularly effective in maintaining pepsin activity when compared to 5% trehalose. In the presence of 10% maltodextrin, the mixture of 2.5% glycerol and 2.5% trehalose showed similar relative activity to 5% glycerol alone, while the mixture of 1.25%% glycerol and 1.25% trehalose exhibited the equivalent relative activity to 2.5% trehalose alone (*p* > 0.05). The results suggest that freeze-drying could lower pepsin activity to a high extent. However, the addition of 10% maltodextrin along with 5% trehalose could enhance the stability of skipjack tuna pepsin or pepsinogen during freeze-drying, in which a relative activity of 80% was attained.

### 3.2. Characteristics of Freeze-Dried Tuna Pepsin Powder Containing 10% Maltodextrin and 5% Trehalose (TPP-T5)

#### 3.2.1. Microstructure

Microstructure of TPP-T5 is shown in Figure 3. The SEM image showed the irregular flake-like particles of TPP-T5 with different sizes. In general, freeze-drying or lyophilization removes water from a frozen sample, through a vacuum system called ‘sublimation’ [24]. Non-uniform grinding mainly caused the heterogeneous shape and size of TPP-T5. However, Caparino et al. [25] reported that the freeze-dried mango powder appeared as smooth flakes. Ismail et al. [26] prepared freeze-dried yogurt with the addition of some additives (spirulina powder, modified starch, and whey protein concentrate). After completing freeze-drying, the resultant product had a flake-like structure. The results indicated that freeze-drying of the maltodextrin-trehalose mixture containing tuna pepsin resulted in flake-like particles.

#### 3.2.2. Particle Size, Poly-Dispersity Index (PDI), and Zeta (ζ) Potential

The particle size of TPP-T5 particles varied between 7.82 and 123.02 μm with a mean size of 65.42 ± 57.60 μm (Table 1). The obtained results were in accordance with SEM images of TPP-T5, in which particles with varying sizes were observed. Particle size is one of the most important physical parameters of powders [27]. Bailey and Cho [28] documented that the specific activity of immobilized glucoamylase increased as particle size decreased. Moreover, particle size and dispersion had a crucial role in colloidal stability, solubility, release, and bioavailability [29].

PDI values of TPP-T5 ranged from 0.416 to 0.528, indicating a moderate size distribution for the powder (Table 1). The size distribution or heterogeneity of the particles in a system is indicated by the PDI. A sample with very wide particle size distribution has a PDI value more than 0.7 and the value less than 0.05 corresponds to a very monodisperse sample [30]. The PDI of TPP-T5 was 0.474, which was higher than 0.05. This coincided with the wide range of particle sizes. Agustinisari et al. [31] documented PDI values of 0.468 to 0.705 for spray-dried whey protein-maltodextrin microcapsules loaded with eugenol.

Ζ-potential of TPP-T5 showed a negative charge (−14.34 mV to −23.84 mV) (Table 1). Ζ-potential values, which typically range from +100 to −100 mV, can be used to predict colloidal stability [30]. An average ζ-potential of TPP-T5 was −19.95 mV, which suggested good colloidal stability. High dispersion stability was commonly noted at ζ-potential with values of <−25 mV or >+25 mV [32]. Lower ζ-potential is associated with coagulation, aggregation, or flocculation, which are caused by Van der Waals inter-particle attraction [33]. In general, a high value of ζ-potential, either negative or positive, indicates the higher stability regulated by the repulsion between the charged surface of macromolecules. Ζ-potential for maltodextrin at 10% was −7.04 mV as documented by Albert et al. [34]. Fonte et al. [35] stated that an insulin-loaded poly(lactic-co-glycolic acid) nanoparticles sample containing 10% trehalose showed a higher negative value of ζ-potential (−42.9 ± 1.7 mV). The results suggested that the high ζ-potential of TPP-T5 was probably due to the partial adsorption of trehalose and maltodextrin on pepsin or pepsinogen. As a result, the negative charge of the enzyme was still dominant when examined.

#### 3.2.3. Density and Flowability

The mass of powders in a particular volume of space is referred to as bulk density (untapped) and is inversely related to the volume of powders. At a given mass, powder occupies less space when the bulk density is higher [36]. The TPP-T5 showed a bulk density of 0.53 kg m^−3^, which might have been due to the irregular or heterogeneous particles and varying sizes of the powders. Teo et al. [37] reported that the spray dried corn oil powder produced with 10% trehalose and maltodextrin showed a bulk density of 0.23 g/mL. The bulk density is influenced by several variables, such as flowability, agglomeration, particle morphology (shape and size), and particle density. Generally, smaller particles have greater bulk densities as they can fit into less area and thus particles pack densely. As a consequence, powders have higher densities. TPP-T5 was flake-like with varying sizes and shapes, resulting in differences in untapped and tapped densities (Table 1). The difference between tapped and untapped densities indicated the heterogenicity of TPP-T5 particles (Figure 3). 

Carr index (Ci) is one of the indices employed in several industries to determine the flowability of powders [16]. Ci was used to describe the flowability of TPP-T5, which was 34.76. The flowability of powders with Ci in the range of 20 to 35 is regarded as fair [16]. The results indicated that TPP-T5 possessed fair flowability, as characterized by Carr index values of 20–35.

#### 3.2.4. Wettability and Solubility

Reconstitution time is mainly related to the agglomeration and the wettability of individual powder upon rehydration. Higher reconstitution time of powders indicated less wettability of powders [16]. Wettability time of TPP-T5 was 16.36 ± 0.73 min. During freeze-drying of TPP-T5, trehalose was possibly associated tightly with water molecules. As result, there were fewer sites for a water molecule to attach upon rehydration. Therefore, a longer time was required for rehydration. Wettability of shrimp oil nanoliposomes freeze-dried powder (14.1 to 15.8 min) was greater than spray-dried powder (20.7 to 22.1 min), due to its larger size [17]. Moreover, powder reconstitution time can vary from a few seconds to hours, depending on the presence of hydrophilic/hydrophobic groups, porosity, structure, etc. [38].

Solubility of TPP-T5 was 93.58% (Table 1). Hydrophobic and hydrophilic components of the dried powder determined its solubility [37]. As a key indicator of the overall quality of powders, solubility has been thought to be important. Saquib and Benjakul [17] reported that freeze-dried powder and spray-dried powder of shrimp oil nanoliposomes samples with high carboxy methyl cellulose proportion had higher solubility (69.1% to 85.22%). Maltodextrin is a water-soluble polysaccharide used as an emulsion stabilizer and tablet binder [37]. The results reflected that maltodextrin and trehalose were hydrophilic compounds and could facilitate the high solubility of TPP-T5. As a consequence, entrapped pepsin or pepsinogen could be liberated easily.

### 3.3. Storage Stability of TPP-T5 at Different Relative Humidities (RHs)

Storage stability of TPP-T5 at different RHs is displayed in Figure 4. Slight loss of pepsin activity was noted in all samples stored at different RHs with increasing storage time (*p* < 0.05). The decrease in relative pepsin activity of TPP-T5 at all RHs used suggested that the degradation or autolysis of pepsin might occur. At week 1, the highest relative pepsin activity was noted for the sample stored in the lowest RH (11%), when compared to other RHs. However, it was significantly decreased at week 4, compared to that found at week 1. A sample stored at RH 11% and RH 33% showed higher relative pepsin activity than those stored at RH 75% and RH 97%, particularly at longer storage time (4 weeks). At high RH, autolysis could be enhanced due to the presence of more free water involved in hydrolysis of enzyme itself. Kawai and Suzuki [39] reported that the storage stability of freeze-dried proteins varied with temperature and RH. Trehalose is the most suitable stabilizer than other disaccharides (sucrose, maltose, and lactose) [39]. The results suggested that 5% trehalose along with 10% maltodextrin played an essential role in the stabilization of enzymes during the extended storage time. Nevertheless, to enhance the stability of pepsin or pepsinogen in TPP-T5, the dry condition with low RH was recommended.

### 3.4. Characterization of Tuna Pepsin (TP) and Commercial Pepsin (CP)

TPP-T5 was used as the source of tuna pepsin (TP), while some inactivated form (pepsinogen) was included. 

#### 3.4.1. SDS-PAGE Protein Pattern

Electrophoretic patterns of TP and CP are displayed in Figure 5a. Different electrophoretic patterns were observed between TP and CP. A clear band with a molecular weight (MW) of 35.2 kDa was detected in CP, representing the purity of the enzyme. The MW of porcine pepsin is 34.6 kDa [40]. The results suggested that the band with MW of 35.2 kDa in CP belonged to pepsin. In addition, another band with low MW was found. This might have been the protein used for stabilizing porcine pepsin in powder form. In TP, several bands with MW of 66 kDa, 43.3 kDa, 37.5 kDa, 32.5 kDa, 26.2 kDa, and 6.7 kDa were observed. Bougatef et al. [41] reported that the purified pepsinogen from the stomach of smooth hound (*Mustelus mustelus*) had a MW of ≈40 kDa. Nalinanon et al. [8] documented that MW of skipjack tuna (*Katsuwonus pelamis*) pepsins 1 and 2 were 33.9 and 33.7 kDa, respectively. The results suggested that crude tuna pepsin contained both pepsinogen and pepsin, as well as other contaminating proteins.

#### 3.4.2. Molecular Weight Distribution

Elution profile of TP and CP by gel filtration is shown in Figure 5b. Two distinct peaks of A280 were observed in CP, indicating the presence of proteins containing aromatic amino acids, with high and low MW. For CP, the MW of peak 1 was 35.4 kDa, which coincided with that found in the electrophoretic pattern of CP (35.6 kDa). On the basis of proteolytic activity, it was reconfirmed that peak 1 with MW of 35.6 kDa represented pepsin. However, peak 2 of CP showed no proteolytic activity. The results suggested that CP plausibly contained the stabilizing protein with low MW. On the other hand, several peaks were noticed in TP. Peak 1 of TP had a MW of 42.6 kDa. Bougatef et al. [41] reported that the purified pepsinogen from the stomach of smooth hound (*Mustelus mustelus*) had a molecular weight of ≈40 kDa when determined by gel filtration. Pepsinogens are converted into active enzymes under acidic conditions via releasing the activation segments from their NH_2_-terminal regions [42]. The results suggested that tuna pepsinogen was converted to active pepsin under acidic conditions, as witnessed by proteolytic activity. Peak 2 of TP with MW of 33.1 kDa more likely represented the active pepsin with proteolytic activity. The results suggested that some active pepsin was present initially in the stomach before the extraction. Other peaks showed negligible proteolytic activity indicating the presence of contaminant proteins in TP. The results revealed that TP contained mainly pepsinogen and a small portion of active pepsin. Thus, TP should be fully activated under acidic conditions before use.

### 3.5. In Vitro Simulated Gastric Digestion of Different Proteins by TP and CP

#### 3.5.1. Protein Patterns

Protein patterns of samples including whey protein isolate (WPI), kidney bean protein isolate (KBPI), and threadfin bream fish mince (TBFM) after being digested using TP and CP at different levels (5, 10, and 15 units/g protein) are depicted in Figure 6. WPI (undigested sample) has six protein bands, representing α-lactalbumin, β-lactoglobulin, immunoglobulins, bovine serum albumin, bovine lactoferrin, and lactoperoxidase [43] (Figure 6a, lane U). Nevertheless, after enzymatic digestion, those protein bands were degraded or vanished with the coincidental generation of proteins or peptides with lower MW. Protein degradation was obvious as the level of the enzyme increased, especially at the enzyme level of 15 units/g protein. WPI was degraded to a higher extent when hydrolyzed with CP when compared to TP. All protein bands vanished when CP was used, especially at the level of 15 units/g proteins (Figure 6a, lane CP15). On the other hand, for TP at the level of 15 units/g proteins, proteins with MW above 20 kDa was completely degraded with concomitant generation of peptides or proteins with lower MW. However, some protein bands below MW of 20 kDa were still retained after hydrolysis. In general, fish pepsins have low activity toward small peptide substrates [44]. 

Protein patterns of KBPI hydrolyzed by TP and CT were different after hydrolysis using various enzyme levels (Figure 6b). Proteases from various sources differ greatly in their cleavage sites, leading to the formation of peptides/proteins with varying sizes [2,45]. CP and TP obtained from porcine and fish, respectively, might have different catalytic sites that plausibly generated peptides with different MWs. When TP (5 units/g protein) was used, similar bands with lower band intensity were observed to that of the undigested sample (Figure 6b, lane U and 5). Nevertheless, protein bands above 25 kDa MW completely disappeared. Proteins with MW below 20 kDa were retained in both hydrolysates, regardless of enzyme levels. This was probably attributed to the high resistance of those plant proteins against pepsin. Plant proteins might contain phytochemicals such as phenolic compounds, which could inhibit pepsin via cross-linking activity [46]. As a result, lower proteolytic activity was attained.

Conversely, drastic hydrolysis was observed for TBFM after being hydrolyzed with TP and CP (Figure 6c). TBFM (undigested sample) had three major protein bands, representing myosin heavy chain (MHC), actin (AC), and tropomyosin I [47] (Figure 6c, lane U). Nevertheless, the complete disappearance of MHC, AC, and TM bands occurred when hydrolyzed using TP and CP at all levels. Results suggested that both enzymes effectively hydrolyzed TBFM. Fish protein is well known for being easily digestible [48]. TP showed slightly lower hydrolyzing capacity in comparison with CP, as witnessed by the remaining protein bands with MW lower than 30 kDa. The results suggested that fish pepsin from skipjack tuna could be recovered with ease and introduced as a potential source for enzyme supplement to replace commercial bovine or porcine pepsin, which has religious restrictions.

#### 3.5.2. Degree of Hydrolysis (DH)

DHs of different protein samples including WPI, KBPI, and TBFM after being hydrolyzed using CP and TP at different levels are shown in Table 2. Various DHs were observed for all protein samples, depending on the levels of TP and CP used. DH for all protein samples increased with increasing levels of TP and CP used (*p* < 0.05). The results indicated that the cleavage of several peptide bonds occurred in all protein samples. Addition of TP and CP, particularly at a higher level, led to higher hydrolysis of proteins. Moreover, high DH was observed in hydrolysate produced by CP than TP, regardless of protein samples (*p* < 0.05). The results indicated that proteins were more susceptible to hydrolysis by CP than TP. Highest DH was observed in TBFM, compared to other protein samples, particularly at a higher level of enzymes used (15 units/g protein). Conversely, the lowest DH was attained in KBPI among other samples. The obtained results were in accordance with protein patterns (Figure 5), in which higher digestion was observed in TBFM than KBPI. The results suggested that the addition of TP and CP, particularly at higher levels, enhanced the protein hydrolysis as indicated by the increased DH. Compared to CP, TP digested all protein samples to a lower extent. Nevertheless, pepsin from tuna stomach could assist in the digestion of all the proteins to a high degree, especially for those who are lacking digestive enzymes.

#### 3.5.3. Size Distribution

MW of peptides including WPI, KBPI, and TBFM after being digested using TP and CP at 15 units/g protein level are shown in Figure 7. Several peaks were noted in spectra, representing the occurrence of several peptides with different MWs in all samples. For WPI hydrolyzed with CP, the peptide with MW of 1012 Da was dominant, followed by those with MW of 2104 and 2159 Da, respectively. On the other hand, the peptide with MW of 3836 was predominant, followed by peptides with MW of 2104 and 2159 Da for WPI hydrolyzed with TP. This result suggested that the hydrolysis using TP and CP generated peptides with different MWs. CP yielded peptides with lower MW. This coincided with higher DH of proteins hydrolyzed by CP. Similarly, KBPI digested with TP and CP showed different dominant peptides with MW of 1174 and 1495 Da, respectively. However, peptides with lower MW were prevalent for proteins hydrolyzed using CP. TBFM digested with TP and CP generated peptides with MW in the range of 513 to 2115 Da and 450 Da to 1674 Da, respectively. The obtained results were in line with the protein patterns, in which different low MW peptides were observed after hydrolysis of proteins using TP and CP at the level of 15 units/g protein. The MW has a significant role in influencing the functional and biological properties of hydrolysates [49]. Several studies have reported that peptides with low MW exhibited potent antioxidant activity, antitumor activity, ACE inhibitory activity, etc. [15,50,51,52]. The results suggested that TP and CP effectively digested all proteins and generated a number of low MW peptides, which might possess some biological activities. In addition, the small peptides or free amino acids could be absorbed into the body easily.

## 4. Conclusions

Pepsin/pepsinogen powder containing 10% maltodextrin and 5% trehalose could stabilize tuna pepsin and pepsinogen during freeze-drying and had good physical characteristics. Moreover, the tuna pepsin powder was stable at extended storage time, especially at low relative humidity. Nevertheless, the effect of different packaging materials on the storage stability of tuna pepsin powder should be investigated in future studies. Although tuna pepsin had a slightly lower hydrolyzing capacity toward different proteins in comparison with commercial porcine pepsin, it could be extracted from stomach, which was abundant and underutilized. Thus, tuna pepsin was considered as a promising supplement for people depletory of digestive enzymes to replace commercial porcine and bovine pepsin, which may not be acceptable for some consumers related to religion restrictions.

## Figures and Tables

**Figure 1 foods-11-02292-f001:**
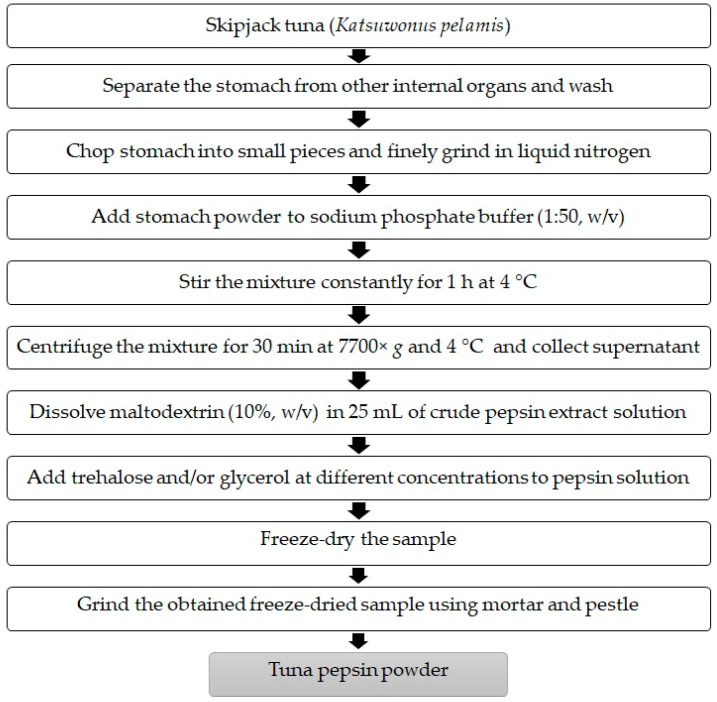
A flow chart of crude tuna pepsin powder production process.

**Figure 2 foods-11-02292-f002:**
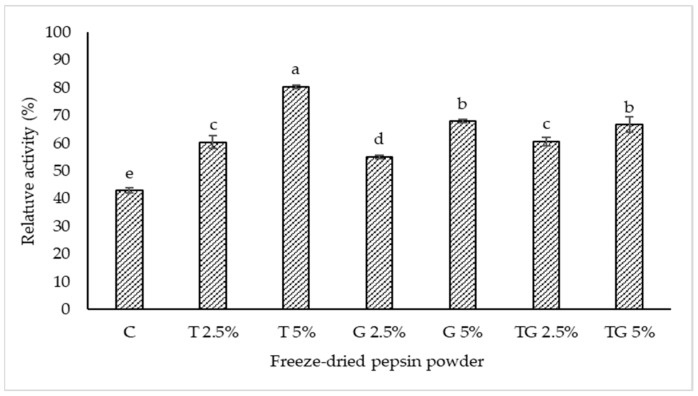
Relative activity of crude tuna pepsin powder containing 10% maltodextrin in combination with trehalose and/or glycerol at different levels after freeze-drying. C—control, T—trehalose, G—glycerol, TG—trehalose/glycerol (1:1). Different lowercase letters on the bars indicate significant differences (*p* < 0.05).

**Figure 3 foods-11-02292-f003:**
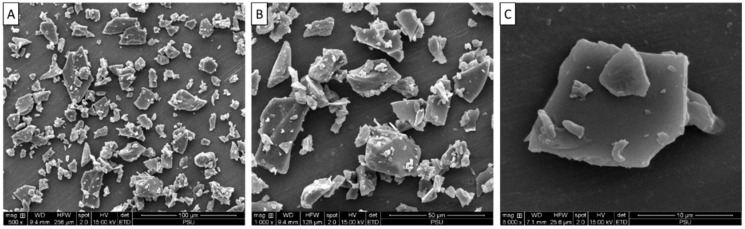
Scanning electron microscopic images of freeze-dried crude tuna pepsin powder containing 10% maltodextrin and 5% trehalose (magnification at ×500 (**A**), ×1000 (**B**), and ×5000 (**C**)).

**Figure 4 foods-11-02292-f004:**
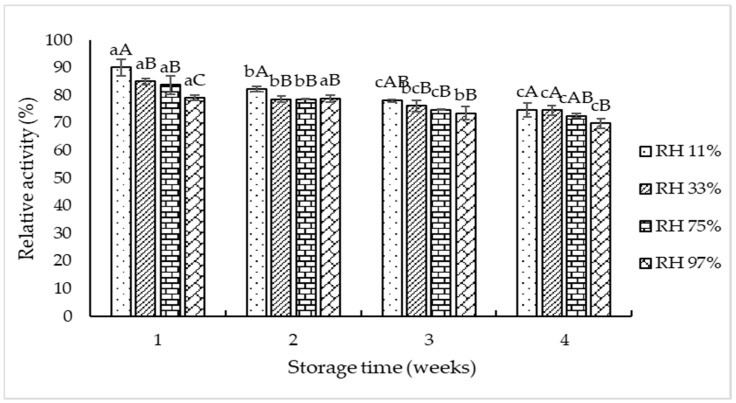
Storage stability of freeze-dried crude tuna pepsin powder containing 10% maltodextrin and 5% trehalose at different relative humidities (RHs). Different lowercase letters on the bars within the same RH indicate significant difference (*p* < 0.05). Different uppercase letters on the bars within the same storage time indicate significant difference (*p* < 0.05).

**Figure 5 foods-11-02292-f005:**
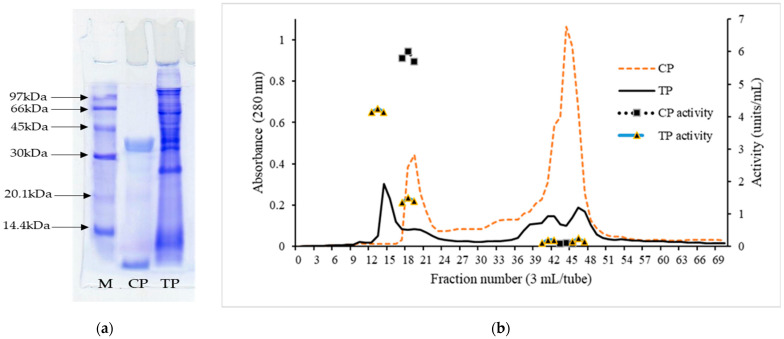
(**a**) Electrophoretic pattern of commercial pepsin (CP) and crude tuna pepsin (TP). Low-molecular-weight marker (M); (**b**) Molecular weight distribution by gel filtration and proteolytic activity of CP and TP.

**Figure 6 foods-11-02292-f006:**
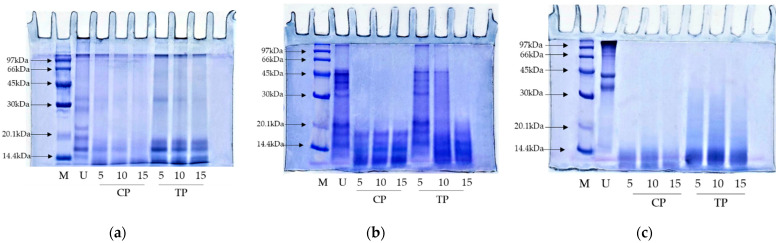
Protein pattern of (**a**) whey protein isolate; (**b**) kidney bean protein isolate; (**c**) threadfin bream fish mincewithout digestion (U) and after being digested using commercial pepsin (CP) and skipjack tuna pepsin (TP) at different levels. Number designates the level of enzyme (units/g protein). Low molecular weight marker (M).

**Figure 7 foods-11-02292-f007:**
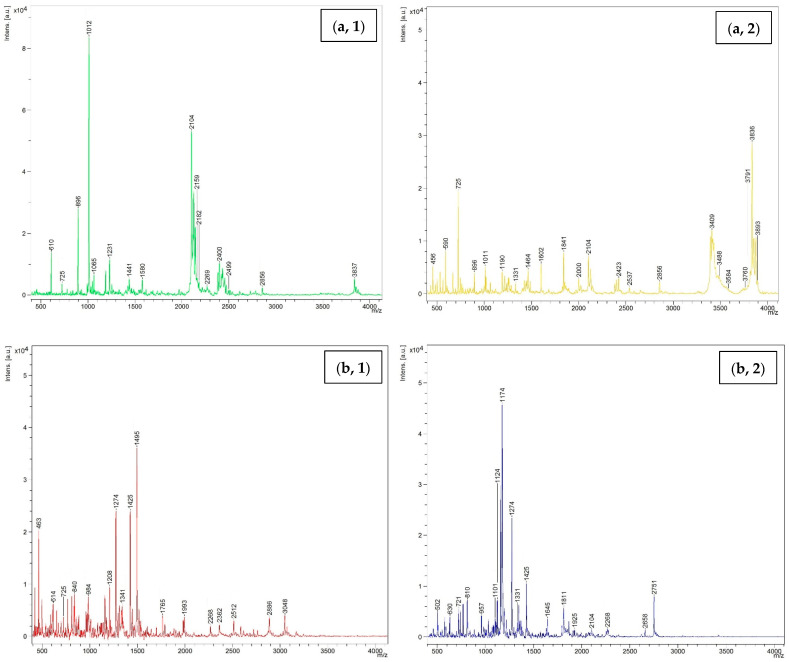
Size distribution of hydrolyzed proteins including (**a**) whey protein isolate; (**b**) kidney bean protein isolate; (**c**) threadfin bream fish mince using commercial pepsin (**1**) and tuna pepsin (**2**) at a level of 15 units/g protein.

**Table 1 foods-11-02292-t001:** Physical properties of freeze-dried crude tuna pepsin powder containing 10% maltodextrin and 5% trehalose.

Sr. No.	Physical Properties	Mean ± SD *
1	Particle size (µm)	65.42 ± 57.60
2	Poly dispersion index (PDI)	0.474 ± 0.01
3	Zeta potential (mV)	−19.95 ± 1.00
4	Untapped density (kg m^−3^)	0.52 ± 0.03
5	Tapped density (kg m^−3^)	0.79 ± 0.05
6	Carr index	34.76 ± 0.61
7	Wettability (min)	16.36 ± 0.73
8	Solubility (%)	93.58 ± 1.25

* Values are presented as mean ± SD (*n* = 3).

**Table 2 foods-11-02292-t002:** Degree of hydrolysis of whey protein isolate, kidney bean protein isolate, and threadfin bream fish mince using commercial pepsin and crude tuna pepsin powder containing 10% maltodextrin and 5% trehalose at different levels.

Samples	Enzyme Level (units/g Protein)	Degree of Hydrolysis (%)	
		CP *	TP **
Whey protein isolate (WPI)	5	59.69 ± 1.66 Ac	32.54 ± 0.81 Bc
	10	74.17 ± 3.88 Ab	40.54 ± 1.29 Bb
	15	94.61 ± 0.85 Aa	63.21 ± 0.64 Ba
Kidney bean protein isolate (KBPI)	5	33.71 ± 1.22 Ac	11.97 ± 1.37 Bc
	10	50.13 ± 2.19 Ab	29.10 ± 1.08 Bb
	15	58.59 ± 0.89 Aa	44.25 ± 1.30 Ba
Threadfin bream fish mince (TBFM)	5	69.93 ± 1.40 Ac	65.65 ± 1.29 Bc
	10	80.94 ± 1.06 Ab	72.26 ± 0.69 Bb
	15	96.60 ± 1.51 Aa	83.24 ± 0.97 Ba

* Commercial porcine pepsin; ** freeze-dried tuna pepsin powder containing 10% maltodextrin and 5% trehalose. Values are presented as mean ± SD (*n* = 3). Different uppercase letters in the same row indicate significant difference (*p* < 0.05). Different lowercase letters in the same column indicate significant difference (*p* < 0.05).

## Data Availability

The data presented in this study are available in article.

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
