# Peer review of "Freeze-Dried Tuna Pepsin Powder Stabilized by Some Cryoprotectants: In Vitro Simulated Gastric Digestion toward Different Proteins and Its Storage Stability"

_foods, 2022, doi:10.3390/foods11152292_

Round 1

Reviewer 1 Report

In this manuscript, authors extracted pepsin from Tuna fish and developed a freeze dried formulation using maltodextrin and trehalose as stabilizers. The manuscript was written well and the study was conducted with necessary analysis. Following are my queries and suggestion:

Section 2.9: Procedure for in vitro digestion was not clear.

·       Section 3.2.1. What is the full form of TTP-T5? Freeze dried product will always endup with flake like morphology.

·       Section 3.2.2: Why there is huge deviation in the particle size? Seems authors didn’t made the uniform  powders from the freeze dried material.

·       Line 384: Authors need to complete the sentence “different lowercase letters….”?

·       It will be ideal to compare the properties [relative activity, particle and solubility characteristics, SEM] of tuna pepsin along with the control pepsin.

·       From the SDS page results, there is higher extend of degradation with control pepsin than the Tuna pepsin. Although same units of pepsin were used, why there is difference in the peptide bands? Did authors measured the activity of control pepsin and tuna pepsin and maintained same quantity in the digestion.

·       Line 533: Authors could also detail the comparison of production cost between the commercial pepsin and tuna pepsin.

Author Response

In this manuscript, authors extracted pepsin from Tuna fish and developed a freeze-dried formulation using maltodextrin and trehalose as stabilizers. The manuscript was written well and the study was conducted with necessary analysis. Following are my queries and suggestion:

***** We greatly appreciate the reviewer for the valuable time spent on the comment and suggestion to improve our manuscript. All queries have been responded and the corrections have been made in the revised manuscript.

Section 2.9: Procedure for in vitro digestion was not clear.

***** Thank you for your comment. Section 2.9 has been improved with more details for better understanding.

Section 3.2.1. What is the full form of TTP-T5? Freeze dried product will always end up with flake like morphology.

***** Thank you for your comment. The full form of TPP-T5 is “Tuna pepsin powder containing 10% maltodextrin and 5% trehalose”. It is mentioned in the 3.2 heading.

We agree with the reviewer that freeze-dried products generally have flake-like morphology. Therefore, we have amended that sentence in the revised text. Please see line number 332-333.

Section 3.2.2: Why there is huge deviation in the particle size? Seems authors didn’t made the uniform powders from the freeze dried material.

***** Thank you for the question. Freeze-drying generally yielded the powder with an uneven particle size as indicated by a wide range of sizes. After freeze-drying, the sample was ground using a pestle and mortar. The details have been included in line 137-138 and the Figure 1. In fact, there is no controlled process to provide the same size or shape. Unlike spray-drying, the particle size is reduced by atomization followed by dehydration. Saqib and Benjakul (2020) reported that the size of freeze-dried powder particles of shrimp oil nanoliposomes varied between 4 and 1800 μm with a mean size of 240.8 ± 237.4 μm. Moreover, in the present study, after freeze-drying, we did not blend the tuna pepsin powder using a blender to avoid denaturation of pepsin.

Gulzar, S., & Benjakul, S. (2020). Nanoliposome powder containing shrimp oil increases free flowing behavior and storage stability. European Journal of Lipid Science and Technology, 122(6), 2000049.

Line 384: Authors need to complete the sentence “different lowercase letters….”?

***** Sorry for the mistake. A correction has been made. Please see line number 433.

It will be ideal to compare the properties [relative activity, particle and solubility characteristics, SEM] of tuna pepsin along with the control pepsin.

***** Thank you for the valuable suggestion. In the manuscript, we compared the aforementioned parameters between the tuna pepsin samples with different treatments. Since we did not have any information on stabilizers used in commercial porcine pepsin. Therefore, it is difficult to compare the properties between commercial pepsin and tuna pepsin including particle and solubility characteristics, SEM. However, the activity of both pepsins was measured using haemoglobin as substrate, in which the same units could be used for hydrolysis.

From the SDS page results, there is higher extend of degradation with control pepsin than the Tuna pepsin. Although same units of pepsin were used, why there is difference in the peptide bands? Did authors measured the activity of control pepsin and tuna pepsin and maintained same quantity in the digestion.

***** Thank you for the point raised by the reviewer. We have measured the activity of commercial pepsin and tuna pepsin and maintained the same quantity (unit) for digestion. Proteases from various sources differ greatly in their catalytic and physical properties (Shahidi and Kamil, 2001; Zhao et al., 2011). In the present study, commercial pepsin and tuna pepsin obtained from porcine and fish, respectively, might have different cleavage sites. As a consequence, different peptides/proteins with varying sizes as shown by different bands were observed. To support this statement, the related reference has been added to the revised manuscript. Please see line numbers 495-496.

Shahidi, F., & Kamil, Y. J. (2001). Enzymes from fish and aquatic invertebrates and their application in the food industry. Trends in Food Science & Technology, 12(12), 435-464.

Zhao, L., Budge, S. M., Ghaly, A. E., Brooks, M. S., & Dave, D. (2011). Extraction, purification and characterization of fish pepsin: a critical review. Journal of Food Processing and Technology, 2(6), 1000126.

Line 533: Authors could also detail the comparison of production cost between the commercial pepsin and tuna pepsin.

***** Thank you for the suggestion. Sorry, we did not find any information on commercial pepsin production cost for comparison. Therefore, the sentences regarding the cost avoided.  

Reviewer 2 Report

The authors have investigated the effects of maltodextrin (10%) in combination with trehalose or glycerol at different levels 12 (2.5% and 5%) and their mixture on the stability of freeze-dried pepsin from skipjack tuna stomach. To sum up, the manuscript is well written, and the results were well discussed. Besides, the study presented novel results and beneficial for food industry.

Introduction:

 As a commercial porcine pepsin is available in market, please provide its properties as affected by different processes as mentioned in the literature review.

Material and methods:

In chapter 2.2, please insert a flow chart of the powder production process.

Please include a reference in chapter 2.3.

In chapter 2.5, please justify the selection of the concentrations you used.

Please include a reference of the equation stated in chapter 2.5.1.

Please include a reference of the method used in chapter 2.6.1.

Please provide more description details for the method used in measuring the density and flowability.

Please include a reference in chapter 2.9.

Please provide more description details for the method used in measuring the degree of hydrolysis (DH) of digests and the equation used in the calculation process.

Results

In Chapter 3.3., the authors recommended to store the prepared powder in dry condition with low RH at 30 ºC in order to enhance the stability of pepsin. But, I am suggesting to perform glass transition experiment to determine the suitable temperature for storage. Besides, study the effect of different packaging materials on the storage stability of this powder.

Conclusion:

Please provide the future studies related to the storage of the prepared powder.

Author Response

The authors have investigated the effects of maltodextrin (10%) in combination with trehalose or glycerol at different levels (2.5% and 5%) and their mixture on the stability of freeze-dried pepsin from skipjack tuna stomach. To sum up, the manuscript is well written, and the results were well discussed. Besides, the study presented novel results and beneficial for food industry.

***** We are thankful to the reviewer for the encouraging and positive comments to improve the manuscript. We have considered the various suggestions made by the reviewers and have accordingly corrected in the revised text.  

Introduction:

As a commercial porcine pepsin is available in market, please provide its properties as affected by different processes as mentioned in the literature review.

***** Thank you for your comment. Commercial pepsin production process is confidential. However, to provide the related information as raised by the reviewer. We have provided the aforementioned information in the text. Please see line 45-48.

Some studies on pepsin including porcine and other sources have been carried out. Jurado et al. (2012) extracted pepsin from swine wastes using different extraction methods, which affected the proteolytic activity and yield of pepsin differently. Similarly, the properties of fish pepsin were also influenced by the extraction process (Zhao et al., 2011).

The above information has been included in the text along with the references.

Jurado, E., Vicari, J. M., Lechug, M., & Moya-Ramirez, I. (2012). Pepsin extraction process from swine wastes. Procedia Engineering, 42, 1346-1350.

Zhao, L., Budge, S. M., Ghaly, A. E., Brooks, M. S., & Dave, D. (2011). Extraction, purification and characterization of fish pepsin: a critical review. Journal of Food Processing and Technology, 2(6), 1000126.

Material and methods:

In chapter 2.2, please insert a flow chart of the powder production process.

***** Thank you for the suggestion. A flow chart has been added. Please see Figure 1.

Please include a reference in chapter 2.3.

***** Reference has been added. Line number 116.

In chapter 2.5, please justify the selection of the concentrations you used.

***** Thank you for your comment. Based on preliminary studies, it was found that no significant difference was observed in relative pepsin activity of pepsin powder containing trehalose and/or glycerol at level higher than 5%. Therefore, we selected 2.5 and 5% concentrations to lower the cost and prevent the dilution of pepsin.

Please include a reference of the equation stated in chapter 2.5.1.

***** Reference has been added. Line number 147.

Please include a reference of the method used in chapter 2.6.1.

***** Reference has been added. Line number 165.

Please provide more description details for the method used in measuring the density and flowability.

***** Thank you for the suggestion. Details for the method used for measuring the density and flowability have been added in the revised text. Please see line number 175-189.

Please include a reference in chapter 2.9.

***** Reference has been added. Line number 237.

Please provide more description details for the method used in measuring the degree of hydrolysis (DH) of digests and the equation used in the calculation process.

***** Thank you for the suggestion. Details for the method used for measuring the degree of hydrolysis (DH) of the digest and the equation used have been added in the revised text. Please see line number 258-268.

Results

In Chapter 3.3., the authors recommended to store the prepared powder in dry condition with low RH at 30 ºC in order to enhance the stability of pepsin. But, I am suggesting to perform glass transition experiment to determine the suitable temperature for storage. Besides, study the effect of different packaging materials on the storage stability of this powder.

***** We find this as an interesting and invaluable suggestion. We totally agree that further study on the effect of different packaging materials on the storage stability of tuna pepsin powder and glass transition must be investigated.

Currently, we are investigating the effect of temperature on the storage stability of tuna pepsin powder packed in a hard capsule and further placed in a blister pack. The storage stability of samples at room temperature (25 °C – 28 °C) and refrigerated temperature (4 °C) for up to 8 weeks is studied. Glass transition experiment to determine the suitable temperature for storage will be conducted in that study as per the reviewer’s suggestion. The results will be reported in another manuscript. We are grateful for the insightful suggestion. Thank you so much.

Conclusion:

Please provide the future studies related to the storage of the prepared powder.

***** Thank you for the suggestion. Future studies related to the storage of the prepared powder have been provided in conclusion. Please see line number 583-584.

Reviewer 3 Report

The manuscript deals with freeze-dried tuna pepsin powder stabilized by some cryoprotectants, in vitro simulated gastric digestion toward different proteins and its storage stability.

The English language must be revised.

Please number all equations.

Materials and methods

Why only one powder was developed with freeze-dried crude tuna pepsin containing 10% maltodextrin and 5% trehalose???

Color determination of the powder??

Results and discussion

This section must be improved. The results must be better compared with other studies available in the literature.

Figures- Please remove the legend from inside each figure and add info to the figure caption.

Conclusion

Please do not repeat your results and focus on your main conclusions.

References

Around 26 references have more than 5 years. Please update your list of references.

Author Response

The manuscript deals with freeze-dried tuna pepsin powder stabilized by some cryoprotectants, in vitro simulated gastric digestion toward different proteins and its storage stability.

***** We greatly appreciate the reviewer for the valuable time to improve the manuscript.

 The English language must be revised.

*****Thank you for your comment. We have thoroughly checked for grammatical mistakes using the ‘Grammarly’ software. Corrections have been made in the revised text.

Please number all equations.

***** Thank you for the suggestion. The number has been added to all equations.

Materials and methods

Why only one powder was developed with freeze-dried crude tuna pepsin containing 10% maltodextrin and 5% trehalose???

***** Thank you for your insightful question. The main objective of the present study was to stabilize tuna pepsin using some cryoprotectants during freeze-drying. From the results, it was observed that the highest relative pepsin activity was found in tuna pepsin powder containing 10% maltodextrin and 5% trehalose when compared to other samples. Therefore, tuna pepsin powder containing 10% maltodextrin and 5% trehalose was selected for further study.

Color determination of the powder??

***** Thank you for your question. We did not determine the color of the powder in this study. Currently, we are investigating the effect of temperature on the storage stability of tuna pepsin powder packed in a hard capsule and further placed in a blister pack. The color of the powder will be determined in the aforementioned study as per the reviewer’s suggestion. The results will be reported in another manuscript. Thank you so much for the suggestion.

Results and discussion

This section must be improved. The results must be better compared with other studies available in the literature.

***** Thank you for your comment. All the results had been already discussed along with the relevant references throughout the text. However, other references have been added. Please see line 300, 397-399, 404-407.

Figures- Please remove the legend from inside each figure and add info to the figure caption.

***** Thank you for your suggestion. Corrections have been made in the revised manuscript.

Conclusion

Please do not repeat your results and focus on your main conclusions.

***** Thank you for your comment. A conclusion has been improved and repeated results have been removed. However, the suggested further study on packaging and storage stability has been provided following another reviewer’s guidance.

 References

Around 26 references have more than 5 years. Please update your list of references.

***** Thank you for your comment. The reference list has been updated by removing old references and adding new references. However, some references that have more than 5 years cannot be removed or replaced, especially the references related with the analytical methods, etc. Some references are also necessary to support research discussion.

Round 2

Reviewer 1 Report

Authors made significant modifications in the revised manuscript and incorporated the suggestions.

Reviewer 2 Report

Th authors have done their efforts in modifying their manuscript. So, it can be accepted in current form

Reviewer 3 Report

The manuscript was improved.